# A General Framework for Dynamic Consistent Submodular Maximization

**Paul Dütting** [1]   **Federico Fusco** [2]   **Silvio Lattanzi** [1]   **Ashkan Norouzi-Fard** [1]   **Ola Svensson** [3]
**Morteza Zadimoghaddam** [1]

## Abstract

Consistency is an important property in dynamic submodular maximization and entails maintaining a near-optimal solution at all times, making only a small number of adjustments to the solution in each step. Prior work has explored this question for the insertion-only case, where the algorithm faces a stream of $n$ insertions, and has established lower and upper bounds for the cardinality-constrained version of the problem. We consider this question in the fully dynamic setting, where the stream of operations may contain both insertions and deletions. We develop a general framework for designing algorithms for this setting, and instantiate it to obtain the first constant-factor approximations with sublinear consistency. For cardinality constraints, we propose a $1/2 - O(\varepsilon)$ approximation that is $O(1/\varepsilon^2)$ consistent. For rank-$k$ matroid constraints, we construct a $1/4 - O(\varepsilon)$ approximation to the dynamic optimum that is $O(\log k/\varepsilon^2)$ consistent.

## 1. Introduction

Submodular maximization is a fundamental topic in machine learning, capturing a diverse range of applications such as data summarization (Lin & Bilmes, 2011; Bairi et al., 2015), video analysis (Zheng et al., 2014), sparse reconstruction (Bach, 2010; Das & Kempe, 2011), and active learning (Golovin & Krause, 2011; Amanatidis et al., 2020). In many of the aforementioned applications, one has to deal with evolving datasets, where data is inserted and removed, and wants to provide a "stable" solution for the end user. For example, in data summarization, it is common to analyze evolving datasets and it is important to have a stable summary under small changes. Similarly, in recommendation systems, one often wants to retrieve suggestions

over an evolving catalog and ideally would modify recommendations only if substantial changes to the catalog occurs. Motivated by these practical applications and the need of "stable" solutions in many of these applications, a recent line of work has started to explore submodular maximization under *consistency* constraints (Dütting et al., 2024; Dütting et al., 2025c). While prior work has focused on the insertion-only case in the cardinality constrained setting, in this paper we investigate the more realistic fully-dynamic setting, both for the cardinality-constrained case and the more general case of matroid constraints. Our work thus contributes to the rapidly growing literature that studies central optimization problems in machine learning from a consistency perspective (see, e.g., Lattanzi & Vassilvitskii, 2017; Jaghargh et al., 2019; Guo et al., 2021; Cohen-Addad et al., 2022).

**Problem Formulation.** Generalizing the model of consistent submodular maximization of Dütting et al. (2024) and Dütting et al. (2025a), we aim to design consistent approximation algorithms for the fully-dynamic environment, where elements can be inserted or deleted by an oblivious adversary. Our algorithms face a sequence of $n$ update operations, each being either an insertion or a deletion[*]. Given a (monotone) submodular function $f$ defined on these elements, our primary objective is to maintain a high-value feasible solution, which does not change much in each step.

Towards defining these goals more formally, let $\text{ALG}_t$ denote the algorithm's solution after the $t^{th}$ stream operation (either an insertion or a deletion). We denote with $x_t$ the element considered in the $t^{th}$ stream operation and with $X_t$ the set of elements that have been inserted up to operation $t$, and not yet deleted. In particular, we have $\text{ALG}_t \subseteq X_t$. We strive for the following two properties:

- **Approximation:** A (randomized) algorithm is an $\alpha$-*approximation*, for $\alpha \leq 1$, if at each time step $t$, it holds $\mathbb{E}[f(\text{ALG}_t)] \geq \alpha f(\text{OPT}_t)$, where $\text{OPT}_t$ is the optimal solution among the available elements $X_t$.

- **Consistency:** An algorithm is $C$-*consistent*, if

---

[1]Google Research [2]Dept. of Computer, Control and Management Engineering, Sapienza University of Rome [3]EPFL. Correspondence to: Federico Fusco <fuscof@diag.uniroma1.it>.

*Proceedings of the 43rd International Conference on Machine Learning*, Seoul, South Korea. PMLR 306, 2026. Copyright 2026 by the author(s).

[*]For simplicity, we assume that a deleted element *cannot* be re-inserted in the stream; this assumption is without loss of generality, as we can imagine adding multiple copies of the same element.

$|\mathrm{ALG}_t \triangle \mathrm{ALG}_{t-1}| \leq C$ for all $t$[†], where $C$ may be non-constant (e.g., a function of $k$).

As in prior work, we allow the algorithm to be randomized, so that the approximation guarantee may hold in expectation, while the consistency bound has to be enforced realization-wise. In this paper, we measure the changes in solution via the symmetric difference, as opposed to simply counting the new elements inserted as in previous works (Dütting et al., 2024; Dütting et al., 2025a). This definition is more natural when deletions may occur; it is however a simple argument to see that the two definitions are equivalent for monotone submodular functions, up to a multiplicative constant of 2.

**Our Results.** We propose a general framework for designing randomized consistent algorithms for the fully-dynamic setting, and instantiate this framework for two important submodular maximization tasks:

– For cardinality constraints, we devise a poly-time $1/2 - O(\varepsilon)$ approximation that is $O(1/\varepsilon^2)$ consistent.[‡]

– For matroid constraints, we give a poly-time $1/4 - O(\varepsilon)$ approximation that is $O(\log k/\varepsilon^2)$ consistent.

These are the first approximation algorithms in the fully-dynamic setting that exhibit consistency that is sublinear in the cardinality of the solution $k$. Prior work explored consistent submodular maximization in the insertion-only case, focusing on the cardinality-constrained setting. For monotone submodular objectives, there is a tight information-theoretic bound of $2/3$ for randomized strategies and a randomized poly-time $\approx 0.51$-approximation (Dütting et al., 2025a). There is also a deterministic poly-time $\approx 0.382$-approximation, and it is known that no deterministic algorithm (poly-time or not) can do better than $1/2$ (Dütting et al., 2024). All these results apply to constant consistency, but the impossibility results are not affected if we allow for sublinear-in-$k$ many changes. Our algorithm almost matches the state-of-the-art $\approx 0.51$-guarantee for poly-time algorithms, while allowing for both insertions and deletions. For matroid feasibility constraints, there is no specific work on consistency, but the streaming algorithm by Chakrabarti & Kale (2015) maintains a solution that is a $1/4$-approximation and only changes by at most one element after each stream insertion. We match this approximation guarantee in the significantly more challenging fully dynamic setting with insertions and deletions, while incurring a still small logarithmic consistency.

Note, the "technical" gap between cardinality and matroid constraints is not new in the submodular maximization liter-

ature, as the latter poses significant challenges. For instance, while the well known (tight) $1 - 1/e$ approximation for offline submodular maximization with cardinality constraints dates back to the seventies (Nemhauser et al., 1978), it took more than three decades to obtain the same result for matroids (Cualinescu et al., 2011). Similarly, in the semi-streaming setting, while a tight $1/2$ approximation was given in Badanidiyuru et al. (2014), and a matching information-theoretic hardness result is shown in Feldman et al. (2023), the best approximation for matroids is currently $0.3178$ (Feldman et al., 2022) (and no better hardness result is known).

**Technical Overview.** Designing consistent algorithms for submodular maximization is challenging since the optimal solution can change drastically from step to step, while the dynamic solution needs to stay close to the optimum with only a limited number of changes. While this is also true in insertion-only settings, this is amplified in settings with deletions. Let $\mathrm{OPT}_t$ denote the optimal solution at time $t$. In an insertion-only model, the insertion of a new element does not significantly degrade solution quality, provided we can modify $\mathrm{ALG}_t$ via a single operation. Specifically, swapping a new element with a random element in $\mathrm{OPT}_t$ yields a $(1/2 - O(1/k))$-approximate solution: Thus, maintaining a high-quality solution after an insertion is relatively straightforward; in contrast, the deletion-only setting is radically different. If $\mathrm{OPT}_t$ is dominated by a single element, its removal may necessitate the insertion of $O(k)$ new elements to compensate for the lost value. In addition, many algorithms for insertion-only or deletion-only models rely on the monotonicity of the optimal solution's value, this property does not hold in a fully dynamic environment. A single insertion may substantially increase the optimal value, while a deletion can sharply decrease it. These frequent, arbitrary fluctuations prevent the use of standard monotonic arguments and complicate the analysis of the online setting.

A further complication compared to prior work comes from the fact that we aim to design online algorithms under matroid feasibility constraints, limiting which elements can be swapped into the solution. For instance, in a partition matroid, we can only replace an element with someone else from the same subset of the underlying partition. Intuitively, we want to maintain a solution which cannot be affected too much by future insertions and deletions, or that can be "repaired" with a sublinear number of changes.

To address these challenges we combine multiple ideas and introduce new ones. Our initial ingredient comes from the literature on deletion-robust submodular maximization (Zhang et al., 2022), where the task is to keep a representative coreset of elements that contains a good solution even if an adversary deletes a predetermined number of elements. Given the online nature of our problem, we need to be protected against an unknown (and time-varying) num-

---

[†] $A \triangle B = A \setminus B \cup B \setminus A$ denotes the symmetric difference.

[‡] Consistently with the literature, $\varepsilon$ is a small constant precision parameter chosen by the algorithm designer. The $O$-notation simply hides some constant multiplicative factors.

ber of deletions. To this end, we maintain online multiple "guesses" (or robustness levels) and corresponding coresets. Our second ingredient is a randomized scheduling scheme that schedules transition periods for each robustness level, so that the coresets corresponding to various deletion robustness levels are recomputed "often enough", while still allowing for transition from one coreset to another in a consistent way. The third ingredient is a hierarchical structure that combines the different coresets in a single dynamic solution. We distill these ingredients into a general framework, that can then be instantiated for different problems. Instantiating this framework entails specifying two submodular routines, a robust and a non-robust one, specialized to the problem at hand. The choice of these routines determines the approximation and consistency guarantees.

For both general matroids and cardinality constraints, the main challenge then lies in designing the robust submodular routines. For matroids, we combine the SWAPPING algorithm by Chakrabarti & Kale (2015), with the deletion-robustness sampling technique of Zhang et al. (2022), which we extend to handle $O(\log k)$ robustness levels. For cardinality constraints, we design a simple greedy algorithm combined with the sampling technique by Zhang et al. (2022) for a single robustness level.

Analyzing these routines poses significant challenges and requires non-trivial ideas. For matroids, the dynamic solution depends on $O(\log k)$ partial solutions, so we need to make sure that the approximation guarantees are still respected. We do that by ensuring that all the elements in a carefully chosen superset of the available elements are accounted for in the analysis. For cardinality constraints, our analysis does not follow the standard path of the greedy analysis. Instead, it relies on a threshold-based argument to bound the marginal contribution of all the elements not in the solution. Our novel hybrid approach bridges the classic greedy method and threshold based selection, establishing a more effective paradigm for submodular optimization.

**Related Work.** Our problem isat the intersection of three lines of research on submodular maximization: consistent algorithms (so far studied only in insertion-only streams), deletion robustness, and fully-dynamic environments.

We already discussed the recent literature on consistent submodular maximization (Dütting et al., 2024; Dütting et al., 2025a). A related line of work considers online submodular maximization with preemption (Buchbinder et al., 2019; Chan et al., 2018). Here, unlike in our problem, discarded elements cannot be added back into the solution, but it is possible to swap arbitrarily-many fresh elements into the solution. Recent work by Buchbinder et al. (2025) explores a very general approach to obtaining consistency guarantees, but differs from our work in that it considers a different

benchmark (the optimal online algorithm) and aims for a low amortized number of changes (rather than our more practical, worst-case per-step guarantee).

In the deletion robust setting (Mirzasoleiman et al., 2017), the algorithm is given a set $V$ and a parameter $d$, and its goal is to find a small enough summary $W \subseteq V$ that is robust to $d$ deletions, i.e., such that the optimal solution in $W \setminus D$ is a good approximation of the optimal one in $V \setminus D$, for any choice of the $d$ elements in set $D$ by an oblivious adversary. There are mainly two approaches to solve this problem: (i) consider various thresholds and sample u.a.r. elements accordingly, such approach yields constant approximations with $|W| \in \Omega(k + d \log k)$ (Mitrovic et al., 2017; Dütting et al., 2025b), (ii) choosing the elements to swap in with a probability that is inversely proportional to their current marginal contribution (Zhang et al., 2022), obtaining (the optimal) $|W| \in \Omega(k + d)$.

In the fully dynamic literature, the goal is to maintain a good solution while exhibiting a small amortized running time. For monotone objectives with cardinality constraints, the state-of-the-art is a $1/2$-approximation with polylogarithmic amortized running time (Lattanzi et al., 2020; Banihashem et al., 2023), which is tight (Chen & Peng, 2022). For matroid constraints, $1/4$-approximation algorithms are known (Banihashem et al., 2024; Dütting et al., 2025c). The main difference between these papers and ours is that we place less emphasis on running time, focusing instead on ensuring a small number of per-round updates to the solution. In particular, the fully-dynamic algorithms in the literature *do* change the whole solution from time to time.

## 2. Preliminaries

A set function $f : 2^X \to \mathbb{R}$ is monotone if for any $S, T \subseteq X$ with $S \subseteq T$ it holds that $f(S) \leq f(T)$. In addition, $f$ is submodular if for any $S, T \subseteq X$ with $S \subseteq T$ and every $x \in X \setminus T$ it holds that $f(x \mid S) \geq f(x \mid T)$, where for a subset $U \subseteq X$, we use $f(x \mid U) = f(U \cup \{x\}) - f(U)$ to denote the marginal value of element $x$ with respect to the subset $U$. A matroid on ground set $X$ is defined by a family $\mathcal{M}$ of subsets of $X$ (called independent sets) satisfying the three axioms: (1) $\emptyset \in \mathcal{M}$, (2) if $T \in \mathcal{M}$ and $S \subseteq T$ implies $S \in \mathcal{M}$, and (3) if $S, T \in \mathcal{M}$ and $|T| > |S|$ then there exists $x \in T \setminus S$ such that $S + x \in \mathcal{M}$[§]. It can be shown that all maximal independent sets have the same cardinality, which is called the rank $k$ of the matroid (see, e.g., Schrijver, 2003). Cardinality constraints are a special case of matroids (known as $k$-uniform matroids), where $S \subseteq X$ is independent if $|S| \leq k$.

We use $n$ to denote the number of stream operations, indexed

---

[§]For simplicity, we adopt the notation $A + x$ instead of $A \cup \{x\}$; similarly, we use $A - x$ to denote $A \setminus \{x\}$.

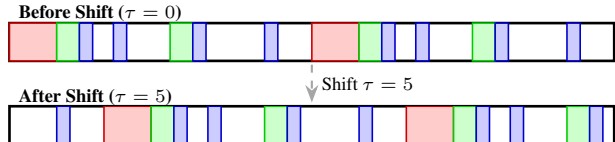

**Before Shift ($\tau = 0$)**

**After Shift ($\tau = 5$)**

Shift $\tau = 5$

*Figure 1.* Visualization of RANDOM-SCHEDULING for $n = 32$, $\ell = 2$, $d_0 = 16$, $d_1 = 8$, $d_2 = 4$, and $\varepsilon = 1/8$. The lower figure corresponds to a possible random output of the routine (resulting from a shift of 5 time steps), while the upper one corresponds to the scheduling before the shift. The transition windows of level 0 are in red, of level 1 are in green, and of level 2 are in blue.

starting from 0 to $n-1$. For simplicity, we assume that such a number is known in advance; however, this assumption can be lifted with minimal adjustments (see Remark 3.1). Following the standard approach in submodular maximization (e.g., Cualinescu et al., 2011), we assume value oracle access to the submodular function (i.e., querying the value of a set $S$ yields $f(S)$) and feasibility oracle access to the matroid constraint (given set $S$, answers whether $S$ is independent or not); The running time of an algorithm is then measured in terms of its calls to these two types of oracles.

## 3. A Consistent Fully-Dynamic Framework

In this Section, we propose a general recipe to design consistent algorithms for fully dynamic submodular maximization, which we then specialize for matroid and cardinality constraints. Our framework is modular and uses three components: a scheduling routine RANDOM-SCHEDULING, a robust submodular maximization routine, and a non-robust one. For now, we use the term "robust" only to differentiate the two submodular routines. All these routines also share a precision parameter $\varepsilon \in (0, 1)$ as a global variable.

At a high level, the framework begins with an initialization phase, performed by RANDOM-SCHEDULING. It receives as input two robustness levels, $d_0$ and $d_\ell$, tailored to the specific problem at hand, and outputs a randomized scheduling. The scheduling algorithm computes a family of robustness levels of the form $d_i = \lceil d_0/2^i \rceil$, interpolating between $d_0$ and $d_\ell$[¶], and specifies transition times for the various robustness levels and corresponding transition windows. The indices of the stream operations are then partitioned into two categories: transition windows and non-transition times. Upon each operation of the stream, the framework does one of two things: if the time index corresponds to a transition time, then it calls the robust routine (corresponding to the given robustness level), and then performs the transition to the newly computed solution during the corresponding transition window. Otherwise (for non-transition times outside

---

[¶]Without loss of generality, we assume that $d_\ell/d_0$ is a power of 2, so that there are $\ell + 1$ robustness levels. Indeed, rescaling $d_\ell$ only affects consistency by at most a constant multiplicative factor.

RANDOM-SCHEDULING

1: **Input:** Initial and target robustness parameters $d_0$ and $d_\ell$, stream length $n$
2: $\mathcal{L} \leftarrow \{d_i = \lfloor d_0/2^i \rfloor \mid i \in \mathbb{Z}_{\geq 0}, d_i \geq d_\ell\}$     {Levels}
3: $\varepsilon' \leftarrow \varepsilon/|\mathcal{L}|$
4: $T_0 \leftarrow [j \cdot d_0 \text{ for } j = 0, \ldots, \lfloor n/d_0 \rfloor - 1]$     {Level 0}
5: $\Delta_0 \leftarrow d_0$
6: **for** $i \in \{1, \ldots, \ell\}$ **do**
7:     $T_i \leftarrow [\cdot]$     {Initialize transitions for level $i$}
8:     **for** each index $t$ in $T_{i-1}$ **do**
9:         Add $t + \varepsilon' d_{i-1}, t + \frac{1}{2}(\varepsilon' d_{i-1} + \Delta_{i-1})$ to $T_i$ if feasible (i.e., if smaller than $n$)
10:         $\Delta_i \leftarrow d_i(1 - i\varepsilon')$
11: Draw a random shift $\tau$ u.a.r. in $\{1, 2, \ldots, n\}$ and add $\tau$ to each index in $T_0, \ldots, T_\ell$, modulo $n$
12: **Return:** $T_0, T_1, \ldots, T_\ell$

transition windows), it calls the non-robust routine. Both routines are passed a suitable temporary solution and a set of candidate elements to process. We describe the three modular ingredients and then detail how they are combined. Note, while the only variability in RANDOM-SCHEDULING entails the input robustness parameters, the submodular routines are problem-dependent.

**Scheduling Routine.** The scheduling routine recursively constructs the transition times corresponding to the various robustness levels. At level 0, it divides the stream into equal chunks of length $d_0$, accounting for a transition window of length $\varepsilon' d_0$ at the beginning of each chunk. Then, it subdivides the remaining $(1-\varepsilon')d_0$ time steps of each chunk in two, accounting for two transition windows of length $\varepsilon' d_1$ each, and so on. This recursive procedure goes on until the last level of robustness is associated with its transition times and corresponding transition windows. Finally, a random shift $\tau$ is chosen u.a.r., and all the time steps are cyclically shifted according to it. For more detailed information, we refer to the pseudocode and to Figure 1. We denote with $T_i$ the set of all the (random) transition times for the generic level $i$ of robustness parameter $d_i$, while the integer interval $[t_i, t_i + \varepsilon' d_i)$ is the corresponding *transition window*.

*Remark* 3.1. For simplicity, we present a version of RANDOM-SCHEDULING that takes as input $n$, but this is not actually needed. It is enough to receive just $d_0$ and $d_\ell$. It can directly compute the deterministic scheduling of the first $d_0$ time steps, perform a cyclic shift drawn u.a.r. in $\{0, 1, \ldots, d_0 - 1\}$, and then repeat itself indefinitely.

The properties enforced by RANDOM-SCHEDULING are presented in the following Lemma, proved in Appendix A.

**Lemma 3.2.** *Any realization of* RANDOM-SCHEDULING *enforces the following properties: (i) The transition windows are non-overlapping. (ii) For any non-transition index*

$t$ and robustness level $i$, the last $t_i$ in $T_i$ before $t$ respects $t - t_i \leq d_i$. If no such $t_i$ exists, then $t < d_i$. (iii) At most an $\varepsilon$ fraction of all times is in a transition window.

**Robust Submodular Routine.** The robust submodular routine, denoted with $\mathcal{A}_R$ in the pseudocode, takes as input three objects: an initial feasible solution, a set of candidate elements to process, and a deletion robustness parameter $d$. At a high level, it modifies the initial solution to account for the high-value elements in the candidate set, while being robust against up to $d$ deletions. The routine then outputs an updated solution $I$ and residual candidate elements $C$.

**Non-Robust Submodular Routine.** The non-robust routine, denoted with $\mathcal{A}_N$ in the pseudocode, simply takes as input an initial solution and a set of candidate elements, and updates the solution according to the set of candidate elements, until they have all been processed. Then, it outputs the updated solution $I$.

**The General Framework.** The general framework combines the three ingredients we have just presented (see also pseudocode). First, it computes the randomized transition times $T_0, \ldots, T_\ell$, according to some initial and target robustness $d_0$ and $d_\ell$. Then, process sequentially the stream:

– **Transition windows.** If the index $t$ of the stream operation is a transition time, i.e., $t \in T_i$ for some robustness level $i$, then the algorithm starts by identifying the last transition time $t'$ for the previous robustness level $T_{i-1}$. Then, it computes $I_t$ and $C_t$ by calling the robust routine $\mathcal{A}_R$ on $I_{t'}$ with candidate set $X_t \setminus (X_{t'} \setminus C_{t'})$ and robustness $d_i$. During the ensuing transition window (operations $t, \ldots, t+\varepsilon' d_i - 1$), the current dynamic solution transitions consistently to $I_t$, removing the elements that get deleted in the meantime.

– **Non-Transition Steps.** If index $t$ of the stream operation does not belong to a transition window, then the algorithm finds the last transition time $t'$ for robustness level $T_\ell$. Then, the solution $\mathrm{ALG}_t$ maintained by the algorithm consists of the output of the non-robust routine on $I_{t'}$ with candidate set $X_t \setminus (X_{t'} \setminus C_{t'})$, after all the elements that are no longer active are deleted.

The candidate sets of the form $X_t \setminus (X_{t'} \setminus C_{t'})$ that are given in input to the submodular routines *do* contain $C_{t'} \cap X_t$. Stated differently, the active elements in the candidate sets output by the routines at the "previous level" are processed by the submodular routines at time $t$. We adopt the convention that if the previous reference time $t'$ is not well defined (because of the scheduling random shift), then the corresponding sets $X_{t'}, I_{t'}$ and $C_{t'}$ are set by default to $\emptyset$.

While it is clear which solution is maintained after the non-transition steps, we still need to specify which solution the algorithm maintains within the transition windows. At a

---

**A Framework for Fully Dynamic Consistent Algorithms**

1: **environment:** Stream $X$, function $f$, matroid $\mathcal{M}$
2: **Input:** Precision $\varepsilon$, robustness parameters $d_0$ and $d_\ell$
3: **Routines:** RANDOM-SCHEDULING, Robust algorithm $\mathcal{A}_R$, non-robust algorithm $\mathcal{A}_N$
4: $T_0, \ldots, T_\ell \leftarrow$ RANDOM-SCHEDULING with precision $\varepsilon$, initial and target robustness $d_0$ and $d_\ell$
5: $t = 0$ \hfill {First time step}
6: **while** $t < n$ **do**
7: \quad **if** $t \in T_i$ for some $i$ **then**
8: \quad\quad Denote with $\mathrm{ALG}_{\mathrm{old}}$ the current solution
9: \quad\quad $t' \leftarrow \max\{\hat{t} : \hat{t} < t, \hat{t} \in T_{i-1}\}$
10: \quad\quad $(I_t, C_t) \leftarrow \mathcal{A}_R(I_{t'}, X_t \setminus (X_{t'} \setminus C_{t'}), d_i)$
11: \quad\quad **for** $j = 0, \ldots \varepsilon' d_i - 1$ **do**
12: \quad\quad\quad $\mathrm{ALG}_t \leftarrow X_t \cap (\mathrm{Shared}_t \cup I_t^{:j} \cup \mathrm{ALG}_{\mathrm{old}}^{j:})$
13: \quad\quad\quad $t \leftarrow t + 1$ \hfill {Next time step}
14: \quad **else**
15: \quad\quad $t' \leftarrow \max\{\hat{t} : \hat{t} < t, \hat{t} \in T_\ell\}$
16: \quad\quad $\mathrm{ALG}_t^T \leftarrow \mathcal{A}_N(I_{t'}, X_t \setminus (X_{t'} \setminus C_{t'}))$
17: \quad\quad $\mathrm{ALG}_t \leftarrow \mathrm{ALG}_t^T \cap X_t$ \hfill {Actual Solution}
18: \quad\quad $t \leftarrow t + 1$ \hfill {Next time step}

---

generic transition time step $t_i$, the algorithm receives the *old* solution $\mathrm{ALG}_{t_i-1}$, and computes the new candidate solution $I_t$ via the robust routine. However, it cannot immediately set $\mathrm{ALG}_{t_i} = I_t$, because this may change too many elements in the solution, thus violating consistency. We note that the common elements $\mathrm{Shared}_t = \mathrm{ALG}_{t_i-1} \cap I_t$ do not need to be touched during the transition window, which then only entails swapping $\mathrm{ALG}_{t_i-1} \setminus I_t$ with $I_t \setminus \mathrm{ALG}_{t_i-1}$, while maintaining feasibility.

The algorithm divides the elements in $I_t \setminus \mathrm{ALG}_{t_i-1}$ into $d_i \varepsilon'$ equal-cardinality chunks $I_t^1, \ldots, I_t^{d_i \varepsilon'}$, and inserts each one of them in one of the ensuing $\varepsilon' d_i$ time steps, swapping out suitable chunks of the old solution $\mathrm{ALG}_{t_i-1} \setminus I_t$: $\mathrm{ALG}_{t_i-1}^1, \ldots, \mathrm{ALG}_{t_i-1}^{d_i \varepsilon'}$. We denote with $I_t^{:j}$ the union $I_t^1 \cup \cdots \cup I_t^j$, and with $\mathrm{ALG}_{t_i-1}^{j:}$ the union $\mathrm{ALG}_{t_i-1}^j \cup \cdots \cup \mathrm{ALG}_{t_i-1}^{d_i \varepsilon'}$, so that the solution maintained by the algorithm in the generic time step $t = t_i + j - 1$ within a time window is $\mathrm{ALG}_t = X_t \cap (\mathrm{Shared}_t \cup I_t^{:j} \cup \mathrm{ALG}_{t_i-1}^{j:})$. In Appendix A, we argue that this can be done while maintaining feasibility of the solution.

## 4. Matroid Constraint

In this Section, we specialize the general framework described to matroid constraints, which we call CONSISTENT-MATROID. To this end, we need to specify the three building blocks. For RANDOM-SCHEDULING, it is enough to set the deletion robustness parameters: $d_0 = k$, and $d_\ell = \log k$, so that the number of robustness levels is $O(\log k)$.

**Algorithm 1** ROBUST-SWAP

1: **input:** Initial solution $I$, candidates $C$, robustness $d$.
2: **while** $|C| \geq d/\varepsilon$ **do**
3:     **for** $u \in C$ **do**
4:         $\Gamma_u \leftarrow \emptyset$
5:         **if** $u+I \notin \mathcal{M}$ **then** $\Gamma_u \leftarrow \{k \in I | I - k + u \in \mathcal{M}\}$
6:         $k_u \leftarrow \arg\min\{w(k) | k \in \Gamma_u\}$
7:     $C \leftarrow \{u \in C | f(u|I) \geq 2w(k_u)\}$
8:     **if** $|C| < d/\varepsilon$ **then break**
9:     Sample $u$ from $C$ w.p. proportional to $1/f(u|I)$
10:    $w(u) \leftarrow f(u \mid I)$ and $I \leftarrow I + u - k_u$
11: **return** $I$ and $C$

**Algorithm 2** SWAP

1: **input:** Initial solution $I$ and candidate elements $C$.
2: **for** $u \in C$ **do**
3:     $\Gamma_u \leftarrow \emptyset$
4:     **if** $u + I \notin \mathcal{M}$ **then** $\Gamma_u \leftarrow \{k \in I | I - k + u \in \mathcal{M}\}$
5:     $k_u \leftarrow \arg\min\{w(k) | k \in \Gamma_u\}$
6:     **if** $f(u|I) \geq 2w(k_u)$ **then**
7:         $w(u) \leftarrow f(u \mid I)$ and $I \leftarrow I + u - k_u$
8: **return** $I$

**ROBUST-SWAP.** The robust routine ROBUST-SWAP starts from an initial solution $I$ and processes the elements in the candidate set $C$ received as input, according to the deletion parameter $d$; see also pseudocode. More in detail, ROBUST-SWAP keeps sampling new elements from $C$ non-uniformly at random in such a way that the probability element $u \in C$ is sampled is inversely proportional to its marginal contribution to the current solution $f(u|I)$. If the sampled element is compatible with the current solution (i.e., $I + u \in \mathcal{M}$), then we simply add it, otherwise we *swap out* a low value element from the solution to make room for it. Before sampling another element, we filter out from $C$ all the elements whose marginal contribution with respect to the current solution is not large enough. As soon as the cardinality of $C$ drops below $d/\varepsilon$, then the routine stops and outputs the new solution $I$, and the set of candidate elements $C$ that have survived the filtering stages but have still not been sampled. Following Chakrabarti & Kale (2015), we introduce the notion of weight $w$ of an element added to the solution, as its marginal contribution upon insertion. We assume that when we pass the initial solution $I$ as input, each element maintains its weight from previous levels.

**SWAP.** The non-robust routine is similar to the robust one, with the crucial difference that it processes *all* the candidate elements $C$ in an arbitrary ordering, without resorting to sampling or anticipated stopping. When the generic $u \in C$ is processed, it is swapped in the solution only if its marginal contribution is at least twice that of the element it replaces. We refer to the pseudocode for the further details.

### 4.1. Analysis of CONSISTENTMATROID

We analyze the matroid algorithm. To this end, we introduce a crucial notion: the sequence of relevant times. The solution in any non-transition time can be seen as the output of a non-robust computation, and a sequence of robust ones, one for each level of robustness. Consider a generic index $t$ that does not correspond to a transition time (i.e., does not belong to any $T_i$ or transition window), the relevant times

are defined recursively starting from $t_\ell$, which is the last transition time in $T_\ell$ before $t$. The generic $t_i$ is then the last transition time in $T_i$ before $t_{i+1}$, and so on, up to $t_0$. For each $i > 0$, CONSISTENTMATROID has built the solution $I_{t_i}$ and candidate set $C_{t_i}$ with a call to ROBUST-SWAP starting with $I_{t_{i-1}}$ and $C_{t_{i-1}}$. An important observation: Once we compute an intermediate set $I_{t_i}$, we do not explicitly delete any element from it in the higher levels $t_{i+1}, \ldots, t_\ell$. The solution $\text{ALG}_t$ is maintained feasible by removing the deleted elements *after* calling SWAP.

We relate the value of the actual solution maintained by the algorithm $\text{ALG}_t$, with the temporary one $\text{ALG}_t^T$ *before* removing the deleted elements. This result tells us that the layered sampling according to the inverse of the marginal contribution is indeed *robust* with respect to the adversarial deletions performed by the stream. As a proxy for the value, we consider the weight function $w(\cdot)$, which is the linear extension of the weights of the single elements, computed upon insertion by ROBUST-SWAP and SWAP.

**Lemma 4.1.** *At a non-transition time $t$, the following holds:* $\mathbb{E}\left[w(\text{ALG}_t)\right] \geq (1 - 8\varepsilon)\mathbb{E}\left[w(\text{ALG}_t^T)\right].$

*Proof.* The temporary solution $\text{ALG}_t^T$ is the outcome of $\ell + 1$ iterative calls (forming a chain similar to composite functions) to ROBUST-SWAP followed by one call to SWAP. Then $\text{ALG}_t$ is computed by simply removing all the elements in $\text{ALG}_t^T$ that are no longer active. Some elements that are added to an intermediate solution $I_{t_i}$ may be swapped out at the same level or in future iterations. Let $U$ be the set of all elements that were ever added to one of these intermediate solutions $I_{t_i}$ or to the solution during the call to SWAP. We start by arguing that this superset $U$ of $\text{ALG}_t^t$ is robust against the deletions, lower bounding $\mathbb{E}\left[w(U')\right]$ in terms of $\mathbb{E}\left[w(U)\right]$ where $U'$ is defined to be $U \cap X_t$. We have the following claim.

**Claim 4.2.** *It holds that $\mathbb{E}\left[w(U')\right] \geq (1 - 4\varepsilon)\mathbb{E}\left[w(U)\right]$.*

The proof is deferred to Appendix A. Let $K$ be the set of elements in $U$ that were, at some point, swapped out by the algorithm to make room for a more valuable one, and denote with $K'$ the elements in $K$ that were not deleted

by the stream $K' = K \cap X_t$. With a simple telescopic argument, similar to Chakrabarti & Kale (2015), we can argue that the weight of $K$ is upper bounded by that of the dynamic solution. The proof is deferred to Appendix A.

**Claim 4.3.** *For any realization of the random sampling, it holds:* $w(K) \leq w(\text{ALG}_t^T)$

We then have the following chain of inequalities that concludes the proof of the Lemma:

$$
\begin{aligned}
&\mathbb{E}\left[w(\text{ALG}_t)\right] \\
&= \mathbb{E}\left[w(U') - w(K')\right] && (w \text{ is linear}) \\
&\geq (1-4\varepsilon)\mathbb{E}\left[w(U)\right] - \mathbb{E}\left[w(K')\right] && (\text{By Claim 4.2}) \\
&= (1-4\varepsilon)\mathbb{E}\left[w(\text{ALG}_t^T) + w(K)\right] - \mathbb{E}\left[w(K')\right] \\
&&& (\text{As } U = \text{ALG}_t^T \cup K) \\
&\geq (1-4\varepsilon)\mathbb{E}\left[w(\text{ALG}_t^T)\right] - 4\varepsilon\mathbb{E}\left[w(K)\right] && (\text{As } K' \subseteq K) \\
&\geq (1-8\varepsilon)\mathbb{E}\left[w(\text{ALG}_t^T)\right],
\end{aligned}
$$

where the last inequality follows from Claim 4.3. $\square$

We are ready to prove the desired approximation guarantee for non-transition times.

**Lemma 4.4.** *At any non-transition time $t$, the following holds:* $\mathbb{E}\left[f(\text{ALG}_t)\right] \geq \left(1/4 - 2\varepsilon\right) f(\text{OPT}_t)$.

*Proof.* The solution at any non-transition time $t$ depends on the call of SWAP at that time, and on the recursive solutions computed by the last calls of ROBUST-SWAP for the various levels of deletion robustness. To analyze the quality of $\text{ALG}_t$, we introduce a superset $\hat{X}$ of $X_t$, composed of all the elements that were, at some point, processed by the algorithm in computing $\text{ALG}_t$. $\hat{X}$ is the union of $X_{t_0} \setminus C_{t_0}$ (i.e., the elements processed al level 0), of $X_{t_1} \setminus (X_{t_0} \cup C_{t_1})$ (i.e., the elements processed at level 1), ... up to $X_{t_\ell} \setminus (X_{t_{\ell-1}} \cup C_{t_\ell})$ (i.e., the elements processed at level $\ell$). Finally, $\hat{X}$ contains the elements processed by the last call of SWAP: $(X_t \setminus X_{t_\ell}) \cup (C_{t_\ell} \cap X_t)$. We denote with $O$ the best independent set in $\hat{X}$. We have that $X_t$ is contained in $(X_{t_\ell} \setminus C_{t_\ell}) \cup (X_t \setminus X_{t_\ell}) \cup (C_{t_\ell} \cap X_t) \subseteq \hat{X}$, therefore $f(O) \geq f(\text{OPT}_t)$, so we aim at upper-bounding $f(O)$.

We have the following technical Claim, which relate the value of $\text{ALG}_t$, $\text{ALG}_T^T$ and of the optimal solution, and whose proof that can be found in Appendix A.

**Claim 4.5.** *The following inequalities hold true:* (i) $f(\text{ALG}_t) \geq w(\text{ALG}_t)$, (ii) $4 \cdot w(\text{ALG}_t^T) \geq f(O)$

We can then conclude the proof of the lemma:

$$
\begin{aligned}
\mathbb{E}\left[f(\text{ALG}_t)\right] &\geq \mathbb{E}\left[w(\text{ALG}_t)\right] && (\text{By (i) of Claim 4.5}) \\
&\geq (1-8\varepsilon)\mathbb{E}\left[w(\text{ALG}_t^T)\right] && (\text{By Lemma 4.1})
\end{aligned}
$$

$$
\begin{aligned}
&\geq (1-8\varepsilon)\frac{f(O)}{4} && (\text{By (ii) of Claim 4.5}) \\
&\geq (1-8\varepsilon)\frac{f(\text{OPT}_t)}{4} && \square
\end{aligned}
$$

**Lemma 4.6.** *The algorithm's consistency is $O(\log k/\varepsilon^2)$.*

*Proof.* To establish the consistency guarantee of CONSISTENTMATROID, we analyze the changes during transition periods and non-transition periods separately. We start by observing that the routines ROBUST-SWAP or SWAP decreases by at least 1 the cardinality of the candidate set every time that they add a new element to the solution.

**Claim 4.7** (Stability of the routines). *Consider any initial solution $I$ and candidate set $C$. The outputs of SWAP and ROBUST-SWAP on $I$ and $C$ are $2|C|$-consistent with respect to the initial solution.*

Consider a generic stream operation $t$, we have two cases.

**Case 1: Non-Transition Periods.** At a time $t$ that is not within any transition window, the algorithm computes the solution $\text{ALG}_t$ by running SWAP. This is initialized with the solution $I_{t'}$ from the last transition time $t' \in T_\ell$, and the candidate set of elements $(X_t \setminus X_{t'}) \cup (C_{t'} \cap X_t)$. We now have two cases.

– If the previous time step was a transition period corresponding to level $\ell$. More precisely, it was the last time step of a time window of length $\varepsilon' d_\ell$. The solution $\text{ALG}_{t-1} = I_{t'}$. We have that $|C_{t'}| \leq d_\ell/\varepsilon$ and $X_t \setminus X_{t'} \leq \varepsilon' d_\ell$, therefore at most $O(d_\ell/\varepsilon) = O(\log k/\varepsilon)$ elements can change during the call of SWAP at time $t$, by Claim 4.7

– If the previous step is also a non-transition period, then $\text{ALG}_{t-1}$ is the result of SWAP on initial set $I_{t'}$ and candidate set $(X_{t-1}\setminus X_{t'}) \cup (C_{t'} \cap X_{t-1})$. The last transition $t'$ is at distance $d_\ell = \log k$ (Lemma 3.2, point (ii)), and $|C_{t'}|$ has cardinality at most $d_\ell/\varepsilon$; therefore $(X_{t-1} \setminus X_{t'}) \cup (C_{t'} \cap X_{t-1})$ has cardinality at most $O(\log k/\varepsilon)$. We thus have the following (by Claim 4.7): $|\text{ALG}_t \triangle \text{ALG}_{t-1}|$ is upper bounded by $|\text{ALG}_t \triangle I_{t'}| + |I_{t'} \triangle \text{ALG}_{t-1}| \in O\left(\frac{\log k}{\varepsilon}\right)$.

**Case 2: Transition Periods.** Consider the case in which $t$ belongs to a transition period. The corresponding period starts at $t_i \in T_i$ and has duration $\varepsilon' d_i$, where $\varepsilon' = O(\varepsilon/\log k)$. During this period, the algorithm gradually transitions from the previous solution, $\text{ALG}_{t_i-1}$, to the newly computed $d_i$-robust solution, $I_{t_i}$, which is the output of ROBUST-SWAP on initial set $I_{t_{i-1}}$ and candidate set $(X_{t_i} \setminus X_{t_{i-1}}) \cup (C_{t_{i-1}} \cap X_{t_i})$. Denote with $t_{i-1}$ the last time that the level $i-1$ has been recomputed, and let $I_{t_{i-1}}$ be the output of that call of ROBUST-SWAP at level $i-1$ (if $t_{i-1}$ is not well defined, then this is the empty set). We have two cases

– If the time step before $t_i$ is a transition period, then it corresponds to level $i-1$, therefore $\text{ALG}_{t_i-1} = I_{t_{i-1}}$, by Claim 4.7, the design of $C'$ and Lemma 3.2 point (ii), we have the following:

$$|\text{ALG}_{t_i-1} \triangle I_{t_i}| \in O\left(|(X_{t_i} \setminus X_{t_{i-1}}) \cup (C_{t_{i-1}} \cap X_{t_i})|\right)$$
$$\in O\left(\frac{d_{i-1}}{\varepsilon}\right).$$

During the $\varepsilon' d_i \in O(d_i \cdot \varepsilon/\log k)$ time steps in the transition window of $t_i$ we are then moving from $\text{ALG}_{t_i-1}$ to $I_{t_i}$, that differ by at most $O\left(d_{i-1}/\varepsilon\right)$ elements, therefore we need to switch $O(\log k/\varepsilon^2)$ elements in each time step of that time window.

– The other case corresponds to the situation in which the time step before $t_i$ is not a transition time. The solution $\text{ALG}_{t_i-1}$ is then the output of SWAP on a suitable base solution and candidate set of elements. We need to recursively unroll the sequence of changes from $I_{t_{i-1}}$ up to $\text{ALG}_{t_i-1}$. Starting from $\hat{t}_\ell$ (i.e., the last time in $T_\ell$ before $t_i$), then $\hat{t}_{\ell-1}$ (i.e., the last time in $T_{\ell-1}$ before $\hat{t}_\ell$), up to $\hat{t}_i$, which is the transition time at level $i$ used in the construction of $\text{ALG}_{t_i-1}$. Finally, denote with $t_{i-1}$[‖] the last transition time in level $T_{i-1}$. We have the following:

$$|\text{ALG}_{t_i-1} \triangle I_{t_i}| \leq |\text{ALG}_{t_i-1} \triangle I_{\hat{t}_\ell}| + \sum_{j=i+1}^{\ell} |I_{\hat{t}_j} \triangle I_{\hat{t}_{j-1}}|$$
$$+ |I_{\hat{t}_i} \triangle I_{t_{i-1}}| + |I_{t_{i+1}} \triangle I_{t_i}|$$
$$\leq \sum_{j=i}^{\ell} \frac{2}{\varepsilon} O(d_i) \in O\left(\frac{k}{\varepsilon} 2^{-i}\right).$$

During the $\varepsilon' d_i \in O\left((\varepsilon k)/(2^i \log k)\right)$ time steps in the transition window of $t_i$ we are then moving from $\text{ALG}_{t_i-1}$ to $I_{t_i}$, that differ by at most $O\left(k/\varepsilon \cdot 2^i\right)$ elements, therefore we need to switch $O(\log k/\varepsilon^2)$ elements in each time step of that time window. □

**Theorem 4.8.** CONSISTENTMATRIX maintains a $(1/4 - 7\varepsilon)$-approximation of the dynamic optimum and is $O(\log k/\varepsilon^2)$-consistent.

*Proof.* Consider the generic time step $t$, we argue that the solution $\text{ALG}_t$ maintained is, in expectation with respect to the randomness of the algorithm, a $1/4 - O(\varepsilon)$ approximation. First, we argue that $t$ is a non-transition time with probability at most $1 - \varepsilon$. This descends from the fact that there are exactly $(1 - \varepsilon)n$ non-transition time steps computed in RANDOM-SCHEDULING, and that the random shift is chosen uniformly at random. We then have the following

$$\mathbb{E}\left[f(\text{ALG}_t)\right] \geq (1-\varepsilon)\mathbb{E}\left[f(\text{ALG}_t)|t \text{ not transition time}\right]$$

[‖]We adopt the convention that, if $i = 0$, then $I_{t_{i-1}} = \emptyset$.

$$\geq (1-\varepsilon)\left(\tfrac{1}{4} - 2\varepsilon\right) f(\text{OPT}_t) \quad \text{(Lemma 4.4)}$$
$$\geq \left(\tfrac{1}{4} - 3\varepsilon\right) f(\text{OPT}_t).$$

The consistency guarantees follow directly from Lemma 4.6, this concludes the proof. □

**Update time.** We measure complexity via the number of calls to the value and matroid independence oracles. Scheduling operations do not query these oracles, so we focus on the online operations. Omitting lower-order terms, the polynomial dependencies on $n$ and $k$ are as follows. For a generic level $i$, the robust routine is called $\Theta(2^i \cdot n/k)$ times, each associated with a transition window of $\Theta(k/2^i)$ time steps. In a generic call to Robust Swap on a candidate set of cardinality $n_i$, the outer while-loop executes at most $n_i$ times because the candidate set $C$ is only shrinking in each iteration. This entails at most $n_i$ value oracle queries per loop, plus the computation of the $k$ initial marginals, yielding $O(n_i^2 + k)$ value oracle calls. The matroid independence oracle is called at most $O(n_i^2 \cdot k)$ times. Since these recurring costs repeat every $\Omega(k)$ time steps, the amortized value and independence oracle calls of level $i$ per update are respectively $O(n_i^2/k)$ and $O(n_i^2)$. The exponential drop in $n_i$ values as $i$ grows makes the aggregate over all levels be dominated by the first term $n_0 \in \Theta(n)$. The corresponding oracle calls for the non-robust routine are also lower order terms because of the same exponential decay structure in place. So the overall amortized update times are $O(n^2/k)$ and $O(n^2)$ for value and independence oracles, respectively.

## 5. Cardinality Constraint

The adaptation of our framework to cardinality, CONSISTENTCARDINALITY, is simpler than its matroid counterpart as it exploits the special structure of cardinality constraint to achieve stronger approximation and constant consistency.

The most notable difference is that we only need *one* single level of robustness, namely $d_0 = d_\ell = \varepsilon^2 k = d$. The robust routine ROBUST-GREEDY thus takes as input no initial solution (as there is *no* intermediate solution inherited from an upper robustness level), but a set of candidate elements, and the single deletion robustness $d$. It constructs a candidate solution $I$ greedily, one element after the other: as long as there are still more than $d/\varepsilon$ candidate elements with positive marginal contribution, it adds a new element sampling non-uniformly from $C$, so that the probability of $u \in C$ being sampled is inversely proportional to $f(u|I)$. We refer to the pseudocode for details.

By design, the solution computed by ROBUST-GREEDY has cardinality at most $k - 2d/\varepsilon$. The candidate set passed to the non-robust routine has cardinality always at most $2\varepsilon k$, in fact, the $C$ computed in ROBUST-GREEDY contains at most

$d/\varepsilon = \varepsilon k$ elements, while there are at most $\varepsilon k$ insertions between the call of ROBUST-GREEDY and the subsequent call to the non-robust routine (note, infact, that the transition windows for the single robustness level are equally spaced at distance $d$). All in all, the *non-robust routine* is therefore extremely simple: just add all the candidate elements as input to the solution $I$.

We report here the properties of the algorithm, and refer to Section A.1 for a formal proof.

**Theorem 5.1.** CONSISTENTCARDINALITY *with precision parameter* $\varepsilon \in (0, 1/2)$ *maintains a* $1/2 - 3\varepsilon$ *approximation of the dynamic optimum and is* $O(1/\varepsilon^2)$*-consistent.*

---

**Algorithm 3** ROBUST-GREEDY

---
1: **input:** Candidate elements $C$ and robustness $d$
2: $I \leftarrow \emptyset$
3: **for** $k - 2d/\varepsilon$ iterations **do**
4:     $C \leftarrow \{u \in C | f(u|I) > 0\}$
5:     **if** $|C| \leq d/\varepsilon$ **then** break
6:     $C' \leftarrow d/\varepsilon$ elements with largest $f(u|I)$ in $C$
7:     Sample $u \in C'$ w. p. proportional to $1/f(u|I)$
8:     $I \leftarrow I + u$
9: **return** $I, C$

---

**Update time.** We call the robust submodular routine $O(n/k)$ times, with each call entailing $O(n \cdot k)$ value oracle queries. The outer for-loop of ROBUST-GREEDY repeats $O(k)$ times, and each call computes the marginals of all elements. The non-robust routine issues no oracle calls. Thus, the amortized update time is $O(n)$.

## 6. Conclusion and Open Problems

In this paper, we introduce the first constant-factor approximation algorithms for fully dynamic submodular maximization that exhibit sublinear consistency. We achieve this by designing a general template for fully-dynamic consistent algorithms, thus paving the way to addressing several interesting open problems: (1) Is there a constant-factor approximation with constant consistency for matroids? (2) Is it possible to break the $1/2$-approximation for cardinality constraints? (3) Can we get similar results for other important submodular optimization problems under consistency constraints (e.g., non-monotone objectives, knapsack and other constraints)?

## Acknowledgments

The work of FF was supported in part by the MUR PRIN grant 2022EKNE5K (Learning in Markets and Society). The work of OS was supported by the Swiss State Secretariat for Education, Research and Innovation (SERI) under contract number MB22.00054.

## Impact Statement

This paper presents work whose goal is to advance the field of Machine Learning. There are many potential societal consequences of our work, none of which we feel must be specifically highlighted here.

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

# A. Missing Proofs

This Appendix is devoted to providing the missing proofs from the main body. We start by formally explaining how to transition from two solutions withing a transition window, as presented at the end of Section 3.

**More on feasible transition.** We are given two feasible sets: the "new solution" $I_t$, and the "old" one $\text{ALG}_{t_i-1}$, and we have to transition from the latter to the former, using $\varepsilon' d_i$ time steps, *while maintaining feasibility* and spreading the changes uniformly within the time steps. Stated differently, we want to find the following partitions:

- $\text{ALG}_{t_i-1} \setminus I_t = \text{ALG}^1_{t_i-1} \cup \cdots \cup \text{ALG}^{d_i\varepsilon'}_{t_i-1}$, with $|\text{ALG}^1_{t_i-1}| = \cdots = |\text{ALG}^{d_i\varepsilon'}_{t_i-1}| = \frac{|\text{ALG}_{t_i-1} \setminus I_t|}{\varepsilon' d_i}$

- $I_t \setminus \text{ALG}_{t_i-1} = I^1_t \cup \cdots \cup I^{d_i\varepsilon'}_t$, with $|I^1_t| = \cdots = |I^{d_i\varepsilon'}_t| = \frac{|\text{ALG}_{t_i-1} \setminus I_t|}{\varepsilon' d_i}$

Denote with $I^{:j}_t$ the union $I^1_t \cup \cdots \cup I^j_t$, and with $\text{ALG}^{j:}_{t_i-1}$ the union $\text{ALG}^{j+1}_{t_i-1} \cup \cdots \cup \text{ALG}^{d_i\varepsilon'}_{t_i-1}$ for generic $j$. The solution maintained by the algorithm in the generic time step $t = t_i + j - 1$ within a time window is

$$\text{ALG}_t = X_t \cap (\text{Shared}_t \cup I^{:j}_t \cup \text{ALG}^{j:}_{t_i-1}).$$

By the strong exchange property of matroids (e.g., Schrijver, 2003) it is indeed possible to find the partitions so that the solution $\text{ALG}_t$ stays feasible.

**Maintaining a feasible solution.**

**Lemma 3.2.** *Any realization of* RANDOM-SCHEDULING *enforces the following properties: (i) The transition windows are non-overlapping. (ii) For any non-transition index $t$ and robustness level $i$, the last $t_i$ in $T_i$ before $t$ respects $t - t_i \leq d_i$. If no such $t_i$ exists, then $t < d_i$. (iii) At most an $\varepsilon$ fraction of all times is in a transition window.*

*Proof.* We prove the three statements for the time indices *before* the random shift, as they are not affected by it.

The statement (i) is clearly true for level 0, as each transition window contains $\varepsilon' d_0$ times, and two consecutive windows are spaced by $d_0$ indices (remember, $\varepsilon \in (0,1)$). For the generic level $i$, we insert two transition windows of length $\varepsilon' d_i$ equally spaced between a transition windows for level $i-1$, and one for some level $j < i$, posed at distance $\Delta_{i-1} = (1 - (i-1)\varepsilon')d_{i-1}$, so there is no overlap (as $i\varepsilon' \leq \varepsilon \leq 1$).

We move our attention to property (ii) and consider a generic non-transition time step $t$ and robustness level $i$. We study the case $i = 0$ separately, and then address the other levels. The non-transition time $i$ belongs to one of the chunks of length $d_0$ between two consecutive transition times for level 0, therefore, its distance to the previous such transition time is at most $d_0$. Consider now the general level $i$. Index $t$ belongs by construction to an interval between a transition window for level $i - 1$ and one for level $j$, for some $j < i$. By design there are at most $d_{i-1}$ time steps between these two windows, and the interval is divided in two by two transition times of level $i$, so that each non-transition index within has distance at most $d_{i-1}/2 = d_i$. The corner cases for the last indices for generic $T_i$ are analogous.

Consider the last point (iii). We restrict our attention to the first $d_0$ time indices, as the fraction is maintained the same in each integer interval of the type $[jd_0, (j+1)d_0)$. Each transition window for level $i$ contains $\varepsilon' d_i$ indices, and there are $d_0/d_i$ of them. All in all, the number of indices in transition windows within the first $d_0$ time indices is:

$$\sum_{i=0}^{\ell} \frac{d_0}{d_i} \varepsilon' d_i = \sum_{i=0}^{\ell} \varepsilon' d_0 = \varepsilon d_0.$$

This concludes the proof. □

**Claim 4.2.** *It holds that $\mathbb{E}[w(U')] \geq (1 - 4\varepsilon)\mathbb{E}[w(U)]$.*

*Proof of Claim 4.2.* Set $U$ is the union of $\ell + 1$ sets $A_0, A_1, \cdots, A_\ell$ and $B$ defined as follows. $A_i$ denotes all elements added to the intermediate set when algorithm ROBUST-SWAP was constructing $I_{t_i}$, while $B$ is the set of elements that were

ever added to the solution in the call to SWAP. We upper bound the loss due to deletions with respect to each of these $\ell + 2$ components. So we have:

$$w(U) = w(B) + \sum_{i=0}^{\ell} w(A_i), \ w(U') = w(B) + \sum_{i=0}^{\ell} w(A_i')$$

where $A_i' = A_i \cap X_t$. Note, $B$ is already contained in $X_t$ by design of the algorithm, as the set $C$ of candidate elements passed to SWAP only contains active elements $X_t$. To prove Claim 4.2, it suffices to lower bound $w(A_i')$ in terms of $w(A_i)$. More precisely, by linearity of expectation, we have that[**]

$$\mathbb{E}\left[w(A_i')\right] = \mathbb{E}\left[\sum_{v \in A_i} w(v)(1 - \mathbb{1}_{\{v \in D_i\}})\right]$$

$$= \mathbb{E}\left[w(A_i)\right] - \mathbb{E}\left[\sum_{v \in A_i} w(v)\mathbb{1}_{\{v \in D_i\}}\right] \tag{1}$$

where $D_i$ is the set of deletions that are relevant in the definition of $A_i$, namely the ones that appears between time $t_i$ and the current time $t$. Denote with $v_i^j$ the $j$'th element added to $A_i$ where $j \le n$. Note, $n$ provides a uniform upper bound on the number of swaps performed by any call the routines, for simplicity we adopt the convention that element $v_i^j$ is well defined for $j$ going up to $n$, as we can always pretend to add dummy elements without affecting approximation and consistency when the actual algorithm stops). We know that $u$ is sampled from a candidate set $C_i^j$ with probability $p_i^j(u) = \frac{1}{w(u)Z(C_i^j)}$ where the normalizing term $Z(C)$ is $\sum_{v \in C} \frac{1}{w(v)}$. One important nuance in our proof is the notion of different relevant deletion sets for each robustness level.

The RANDOM-SCHEDULING algorithm ensures that between $t_i$ and $t$ in this time period at most $|D_i| \le 2d_i \le k/2^{i-1}$ elements have been deleted (by Lemma 3.2 and definition of the deletion parameters). On the other hand, ROBUST-SWAP ensures that the candidate sets we sample from in level $i$ are always larger than $|C_i^j| \ge d_i/\varepsilon \ge k/\varepsilon 2^{i+1}$. Therefore, for any robustness level $i$, and element $j$ added in that level, it holds that $|D_i| \le 4|C_i^j|\varepsilon$. With these observations in mind, we have the following:

$$\mathbb{E}\left[\sum_{v \in A_i} w(v)\mathbb{1}_{\{v \in D_i\}}\right]$$

$$= \sum_{j \le n} \mathbb{E}\left[\sum_{u \in C_i^j} w(u)p_i^j(u)\mathbb{1}_{\{u \in D_i\}}\right]$$

$$= \sum_{j \le n} \mathbb{E}\left[\sum_{u \in C_i^j} w(u)\frac{1}{w(u)Z(C_i^j)}\mathbb{1}_{\{u \in D_i\}}\right] \qquad \text{(Conditional Expectation and def. of } p_i^j\text{)}$$

$$= \sum_{j \le n} \mathbb{E}\left[\frac{1}{Z(C_i^j)}\sum_{u \in C_i^j}\mathbb{1}_{\{u \in D_i\}}\right] \le \sum_{j \le n}\mathbb{E}\left[\frac{|D_i|}{Z(C_i^j)}\right]$$

$$\le \sum_{j \le n}\mathbb{E}\left[\frac{4\varepsilon|C_i^j|}{Z(C_i^j)}\right] = 4\varepsilon\mathbb{E}\left[w(A_i)\right] \qquad \text{(As } |D_i| \le 4|C_i^j|\varepsilon\text{)}$$

Combining the last inequality with Equation (1) yields Claim 4.2. □

**Claim 4.3.** *For any realization of the random sampling, it holds:* $w(K) \le w(\text{ALG}_t^T)$

*Proof of Claim 4.3.* For the generic element $u \in K$, denote with $s_u$ the element that has been swapped in $I$ to replace $u$. By design of ANY-SWAP, $w(s_u) - w(u) \ge w(u)$, therefore, we have the following expansion of $w(K)$:

$$w(K) \le \sum_{u \in K} w(s_u) - w(u) \le w(\text{ALG}_t^T),$$

---

[**]We use that $\mathbb{1}_{\{A\}}$ is the indicator random variable of event $A$, and value 1 is $A$ is realized, and 0 otherwise.

where the last inequality follows by a telescopic argument. $\qquad\square$

**Claim 4.5.** *The following inequalities hold true:* (i) $f(\text{ALG}_t) \geq w(\text{ALG}_t)$, (ii) $4 \cdot w(\text{ALG}_t^T) \geq f(O)$

*Proof.* We start by proving point (i). Sort the elements in $U$ according to the order in which they are added to the solution, and denote with $I_u$ the solution upon insertion of the generic $u \in U$. Starting from a telescopic decomposition of $\text{ALG}_t$, we have the following:

$$
\begin{aligned}
f(\text{ALG}_t) &= \sum_{u \in \text{ALG}_t} f(u | \text{ALG}_t \cap I_u) \\
&\geq \sum_{u \in \text{ALG}_t} f(u | I_u) \qquad\qquad\qquad\qquad \text{(By submodularity)} \\
&= \sum_{u \in \text{ALG}_t} w(u),
\end{aligned}
$$

where the last equality follows from the definition of $w$.

We now move to point (ii). By submodularity, we have that

$$
\begin{aligned}
f(O) &\leq f(U) + \sum_{o \in O} f(o | U) \\
&\leq w(U) + \sum_{o \in O} f(o | U) \qquad\qquad\qquad \text{(same as Claim 4.5)} \\
&\leq w(\text{ALG}_t^T) + w(K) + \sum_{o \in O} f(o | U) \\
&\leq 2 w(\text{ALG}_t^T) + \sum_{o \in O} f(o | U) \qquad\qquad \text{(By Claim 4.3)}
\end{aligned}
$$

We conclude by arguing that the second term of the last line is upper bounded by $2w(\text{ALG}_t^T)$. This descends by defining a notion of weight for any element in $\hat{X}$, not only for those in $U$. Formally, the weight of an element *not in $U$* is its marginal contribution to the solution at the moment that the element is removed from the candidate set $C$, in one of the $\ell + 1$ calls of ROBUST-SWAP, or during the final SWAP. With this definition, we can apply directly Theorem 1 of Badanidiyuru (2011) (using a single matroid and setting $r = 2$), similarly to what has been done e.g., in the appendix of Dütting et al. (2025b). It is crucial to note that—by design of the algorithm and definition of $\hat{X}$— all the elements in $\hat{X}$ are at some point "processed" by a submodular routine, so that they are all "accounted for". $\qquad\square$

### A.1. Cardinality Constraint

We start by arguing that ROBUST-GREEDY is indeed robust to deletions.

**Lemma A.1.** *At a non-transition time $t$, the following holds:*

$$
\mathbb{E}\left[f(\text{ALG}_t)\right] \geq (1 - 2\varepsilon)\mathbb{E}\left[f(\text{ALG}_t^T)\right]
$$

*Proof.* Fix any realization of RANDOM-SCHEDULING, and consider the generic non-transition time $t$. The current solution $\text{ALG}_t$ is computed starting from the last call of ROBUST-GREEDY (at transition time $t'$) and by adding all the active candidate elements from that call, plus all the elements arrived since then but still not deleted. Denote with $D$ the elements in $X_{t'} \setminus X_t$, i.e., the elements that were active upon the last call of ROBUST-GREEDY but deleted since then. Denote with $I_{t'}$ the solution computed by ROBUST-GREEDY, and denote with $e_j$ the random variable representing the random $j^{th}$ element added during the routine to the temporary solution $I_{t'}^j$. Consider any run of the algorithm up to time $j$ (excluded), so that the current solution $I_{t'}^j$ is deterministically fixed. The actual element $e_j$ is drawn from the deterministic set $C_j'$ that contains the $d/\varepsilon$ elements with largest contribution to $I_{t'}^j$, with probability inversely proportional to their marginal contribution to $I_{t'}^j$. The expected contribution of element $e_j$ to the final solution is only affected by a $(1 - \varepsilon)$ factor from the adversarial deletions:

$$
\mathbb{E}\left[f(e_j | I_{t'}^j) \mathbb{1}_{\{e_j \in D\}}\right]
$$

$$= \sum_{e \in C'_j} f(x|I^j_{t'}) \mathbb{1}_{\{e \in D\}} \mathbb{P}\left(e_j = e\right) \qquad (\mathbb{1}_{\{e \in D\}} \text{ is det.})$$

$$= \sum_{e \in C'_j} f(e|I^j_{t'}) \mathbb{1}_{\{e \in D\}} \frac{1/f(e|I^j_{t'})}{\sum_{y \in C'_i} 1/f(y|I^j_{t'})} \qquad (\text{By def.})$$

$$= \frac{|D \cap C'_j|}{\sum_{y \in C'_j} 1/f(y|I^j_{t'})}$$

$$\leq \frac{d}{\sum_{y \in C'_j} 1/f(y|I^j_{t'})} \qquad (\text{Because } |D \cap C'_j| \leq |D| \leq d)$$

$$= \frac{d}{|C'_j|} \mathbb{E}\left[f(e_j|I^j_{t'})\right]$$

$$= \varepsilon \cdot \mathbb{E}\left[f(e_j|I^j_{t'})\right] \qquad (\text{as } |C'_j| = d/\varepsilon)$$

Since the above inequality holds for any possible run of ROBUST-GREEDY up to time $j$, it also holds in expectation overall by the law of total probability. We have the following chain of inequalities:

$$\mathbb{E}\left[f(I_{t'} \setminus D)\right] = \sum_{j=1}^{k(1-2\varepsilon)} \mathbb{E}\left[\mathbb{1}_{\{e_j \notin D\}} f(e_j|I^j_{t'} \setminus D)\right]$$

$$\geq \sum_{j=1}^{k(1-2\varepsilon)} \mathbb{E}\left[\mathbb{1}_{\{e_j \notin D\}} f(e_j|I^j_{t'})\right]$$

$$= \sum_{j=1}^{k(1-2\varepsilon)} \mathbb{E}\left[(1 - \mathbb{1}_{\{e_j \in D\}}) f(e_j|I^j_{t'})\right]$$

$$= \mathbb{E}\left[f(I_{t'})\right] - \sum_{j=1}^{k(1-2\varepsilon)} \mathbb{E}\left[\mathbb{1}_{\{e_j \in D\}} f(e_j|I^j_{t'})\right]$$

$$\geq \mathbb{E}\left[f(I_{t'})\right] (1 - \varepsilon). \qquad (2)$$

Note, it is possible that ROBUST-GREEDY stops before adding $k(1 - 2\varepsilon)$ elements because there are not enough elements left in $C$. In that case, the analysis still goes through by pretending to add some dummy elements of no value. All in all, we have:

$$\begin{aligned}
&(1 - \varepsilon)\mathbb{E}\left[f(\text{ALG}^T_t)\right] \\
&= (1 - \varepsilon)\mathbb{E}\left[f(I_{t'}) + f(X_t \setminus (X_{t'} \setminus C_{t'})|I_{t'})\right] \\
&\leq \mathbb{E}\left[f(I_{t'} \setminus D) + f(X_t \setminus (X_{t'} \setminus C_{t'})|I_{t'})\right] \qquad (\text{By Equation (2)}) \\
&\leq \mathbb{E}\left[f(I_{t'} \setminus D) + f(X_t \setminus (X_{t'} \setminus C_{t'})|I_{t'} \setminus D)\right] \\
&= \mathbb{E}\left[f(\text{ALG}_t)\right].
\end{aligned}$$

Where the last equality follows by the fact that the non-robust routine for cardinality simply adds all the elements in $(X_t \setminus X_{t'}) \cup C_{t'}$ that are still active. $\qquad \square$

Consider now a generic non-transition time $t$, and denote with $t'$ the last transition time before it. The temporary solution $\text{ALG}^T_t$ is the union of the output $I_{t'}$ of ROBUST-GREEDY at time $t'$, the candidate set $C_{t'}$, and the recent element $X_t \setminus X_{t'}$. We have the following property:

**Lemma A.2.** *There exists a threshold value $\theta$ that verifies the following two properties:*

(i) $f(I_{t'}) \geq |I_{t'}| \cdot \theta$,

(ii) $f(x|\text{ALG}^T_t) \leq \theta$ for all $x \in X_t \setminus \text{ALG}^T_t$.

*Proof.* The desired threshold $\theta$ has a simple formula as a function of the sets $I_{t'}$ and $C_{t'}$ in the output of ROBUST-GREEDY:

$$\theta = \min_{x \in C_{t'}} f(x|I_{t'})$$

Let $\{e_1, \ldots, e_{|I_{t'}|}\}$ be the elements in $I_{t'}$ according to the order in which they were inserted, and denote with $I_{t'}^j$ the prefix of the first $j - 1$ such elements. By submodularity, the quantity $\min_{x \in C} f(x|I_{t'})$ only decreases as the algorithm progresses (because submodularity implies that $f(e|I)$ only decreases when elements are added to $I$). Therefore, $\theta$ is smaller than the marginal contribution of any element added to $I_{t'}$ during the algorithm. Thus, we have the following:

$$f(I_{t'}) = \sum_{j=1}^{|I_{t'}|} f(e_j \mid I_{t'}^j) \geq |I_{t'}| \cdot \theta.$$

This concludes the proof of (i). We move to (ii), for which we need to prove that any element $x$ in $X_t \setminus \text{ALG}_t^T$ has a marginal contribution with respect to $\text{ALG}_t^T$ that is upper bounded by the threshold $\theta$. $\text{ALG}_t^T$ contains the elements in $I_{t'}$, those in $C_{t'}$, and the recent elements $X_t \setminus X_{t'}$, therefore we only need to address the elements that were in $X_{t'}$ but were discarded by ROBUST-GREEDY, i.e. $X_t \setminus I_{t'} \cup C_{t'}$. Such elements were characterized by the fact that their marginal contribution to $I_{t'}$ is smaller than that of all the elements in $C_{t'}$, and thus it is smaller than $\theta$. Recall, $I_{t'} \subseteq \text{ALG}_t^T$, therefore, by submodularity, we then have

$$f(x|\text{ALG}_t^T) \leq f(x|I_{t'}) \leq \theta. \qquad \square$$

**Lemma A.3.** CONSISTENTCARDINALITY *has consistency* $O(1/\varepsilon^2)$.

*Proof.* From one non-transition time to the next, the solution changes by at most one element (either due to a deletion or an addition). Consider now a generic transition window; it contains $\varepsilon^2 k$ time indices, and it has to accommodate at most $k$ changes in the solution, for a per-time-step rate of $O(1/\varepsilon^2)$ changes in the solution. $\qquad \square$

**Theorem 5.1.** CONSISTENTCARDINALITY *with precision parameter* $\varepsilon \in (0, 1/2)$ *maintains a* $1/2 - 3\varepsilon$ *approximation of the dynamic optimum and is* $O(1/\varepsilon^2)$-*consistent.*

*Proof.* We start by addressing the approximation claim. Consider a generic non-transition time $t$, and the last transition time $t'$ before $t$. Denote with $\text{OPT}_t = \{o_1, \ldots, o_k\}$ the optimal dynamic solution at time $t$, and let $\theta$ be the threshold as in Lemma A.2. We have now two cases, depending on the cardinality of $|I_{t'}|$.

If $|I_{t'}| = (1 - 2\varepsilon)k$, then

$$
\begin{aligned}
f(\text{OPT}_t) &\leq f(\text{ALG}_t^T) + f(\text{OPT} \mid \text{ALG}_t^T) \\
&\leq f(\text{ALG}_t^T) + \sum_{i=1}^k f(o_i \mid \text{ALG}_t^T) \\
&\leq f(\text{ALG}_t^T) + k\theta && \text{(By (ii) in Lemma A.2)} \\
&\leq f(\text{ALG}_t^T) + \frac{1}{1 - 2\varepsilon} f(I_{t'}) && \text{(By (i) in Lemma A.2)} \\
&\leq \left(\frac{2 - 2\varepsilon}{1 - 2\varepsilon}\right) f(\text{ALG}_t^T). && (3)
\end{aligned}
$$

If $|I_{t'}| < k(1 - 2\varepsilon)$, then it means that all the elements in $X_{t'} \setminus C_{t'}$ have zero marginal contribution to $I_{t'}$. Since all the elements in $C_{t'} \cup (X_t \setminus X_{t'})$ are added to $\text{ALG}_t^T$, and thus have zero marginal contribution to it. All in all, this implies that all the elements in the optimal solution have zero marginal with respect to the temporary solution:

$$
\begin{aligned}
f(\text{OPT}_t) &\leq f(\text{ALG}_t^T) + f(\text{OPT}_t \mid \text{ALG}_t^T) \\
&\leq f(\text{ALG}_t^T) + \sum_{i=1}^k f(o_i \mid \text{ALG}_t^T) \\
&= f(\text{ALG}_t^T). && (4)
\end{aligned}
$$

Combining Equations (3) and (4) with the robustness bound of Lemma A.1, yields the following inequality for any non-transition time steps $t$:

$$\mathbb{E}\left[f(\mathrm{ALG}_t)\right] \geq \left(\tfrac{1}{2} - 2\varepsilon\right) f(\mathrm{OPT}_t). \tag{5}$$

Note, the expectation is only with respect to the random sampling performed by the calls to ROBUST-GREEDY.

Consider now the generic time step $t$, it belongs to a transition window with probability at most $\varepsilon$ (by Lemma 3.2). All in all, we have:

$$\begin{aligned}
\mathbb{E}\left[f(\mathrm{ALG}_t)\right] &\geq (1-\varepsilon)\mathbb{E}\left[f(\mathrm{ALG}_t)|t \text{ not transition time}\right] \\
&\geq (1-\varepsilon)\left(\tfrac{1}{2}-2\varepsilon\right)f(\mathrm{OPT}_t) &\text{(By Eq. 5)} \\
&\geq \left(\tfrac{1}{2}-3\varepsilon\right)f(\mathrm{OPT}_t).
\end{aligned}$$

The consistency guarantees follow directly from Lemma A.3, this concludes the proof. □

