# OpenReview forum: "A General Framework for Dynamic Consistent Submodular Maximization"
_ICML.cc/2026/Conference — ICML 2026 regular_

### Official Review · Reviewer_QNns · 2026-03-11

**Soundness:** 4
**Presentation:** 4
**Significance:** 4
**Originality:** 4
**Overall Recommendation:** 5
**Confidence:** 2

**Summary:**

This paper proposes algorithms for submodular function maximization under cardinality and matroid constraints, in a fully dynamic online setting: at each time step, items can be inserted or deleted from a dynamically changing universe. The aim is to design algorithms that are provable qualitative (good approx. factor, thus near-optimal) at each step, but also consistent, meaning that the algorithms output should preferably not change too much between each consecutive steps.

It is clear that these objectives can be quite conflicting, especially in the fully dynamic setting, where an item deletions might profoundly change the optimum, which then in turn might lead to inconsistency. To address these issues, the authors combine several ideas.
First they maintain multiple “guesses” (or robustness levels) and corresponding coresets. Secondly, they use a randomized scheduling scheme that schedules transitions between levels and ensures that individual levels are updated frequently enough.
The third ingredient essentially combines different levels into one dynamic solution, using a certain hierarchy between levels.

**Compliance With Llm Reviewing Policy:**

Affirmed.

**Final Justification:**

Initially, I did not have many concerns about this paper. After reading the other reviews, I do agree with some concerns shared by the other reviewers, but overall I decide to keep my score (except lowered my confidence by one point). The authors did a reasonable job explaining those concerns in the rebuttal.

**Key Questions For Authors:**

See above

**Limitations:**

yes

**Strengths And Weaknesses:**

Overall, I think this is a good contribution.

S1) Fully dynamic and consistent submodular maximization is clearly a non-trivial task, yet it is an important problem with numerous applications. The authors do a great job of combining existing and novel ideas in order come up with qualitative algorithms in terms of approximation and consistency.

S2) They detail a general framework, beyond submodular maximization, on how to deal with optimization problems in a fully dynamic setting with consistency requirements. This framework generalization further strengthens the paper.

S3) The paper is written very clearly, the main ideas and proofs are understandable even for non-experts like myself.

--

W1) Maybe an experimental evaluation would have been interesting. But this is purely optional, the theoretical contributions are clearly good enough, in my opinion.

---

> ### Author Rebuttal · Authors · 2026-03-31
>
> We thank the reviewer for their thoughtful comments and suggestions.
>
> Empirical Validation. Please see our reply to the reviewer XVSv.

---

> > ### Author Rebuttal · Reviewer_QNns · 2026-03-31
> >
> > Thanks for the answer regarding empirical evaluation, my concerns have been mostly addressed.

---

### Official Review · Reviewer_XVSv · 2026-03-12

**Soundness:** 3
**Presentation:** 3
**Significance:** 2
**Originality:** 3
**Overall Recommendation:** 4
**Confidence:** 3

**Summary:**

The paper studies the problem of dynamic consistent submodular maximization, where the ground set evolves over time through insertions and deletions, and the algorithm must maintain a near-optimal solution while limiting the number of changes between consecutive solutions consistency constraint

The paper proposes a general framework for designing algorithms in the fully dynamic setting and derives concrete algorithms achieving:
- an approximation ratio of $ \frac{1}{2} - O(\epsilon) $ for cardinality constraints,
- an approximation ratio of $ \frac{1}{4} - O(\epsilon) $ for matroid constraints,

**Compliance With Llm Reviewing Policy:**

Affirmed.

**Final Justification:**

The paper focuses on a theoretically grounded problem and provides meaningful contributions from a theoretical perspective. While the work does not aim at improving classical approximation guarantees, it introduces a novel formulation and offers non-trivial algorithmic insights for submodular maximization in the fully dynamic setting.

After reading the rebuttal, I find the authors’ explanations satisfactory. They clarified the key technical ideas, including the role of consistency and the algorithmic design choices, and addressed most of my concerns. In particular, the positioning of the work within the existing theoretical literature is now clearer.

Overall, I acknowledge the theoretical merit of the paper and am satisfied with the authors’ responses, which have improved my assessment.

**Key Questions For Authors:**

Please explain the points mentioned in the weaknesses.

**Limitations:**

The technical contributions mainly extend standard techniques from submodular optimization to the dynamic consistent setting, resulting in limited algorithmic novelty. In addition, the practical relevance of the framework is unclear due to the assumption that the consistency parameter $C$ can be estimated in advance and the lack of experimental validation.

**Strengths And Weaknesses:**

Strengths

- The paper studies a general variant of dynamic submodular maximization with consistency constraints under both cardinality and matroid constraints.

- The authors propose new algorithms within this framework to maintain approximate solutions while ensuring consistency across updates.

Weaknesses

- The technical analysis mainly relies on standard techniques commonly used in submodular optimization. The proofs largely extend existing analytical techniques to the dynamic consistent setting rather than introducing fundamentally new technical tools. As a result, the level of technical novelty appears somewhat limited.

- The proposed framework relies on the assumption that the consistency parameter $C$ can be estimated or controlled in advance. However, this assumption may be unrealistic in many practical scenarios involving dynamic data. The paper does not clearly explain under what concrete applications such a parameter can be reliably estimated or bounded. Providing examples or discussions of settings where $C$ can be meaningfully determined would strengthen the practical relevance of the framework.

- While the paper employs several algorithmic techniques, it does not provide experimental evaluations to demonstrate their empirical performance. It would be useful to validate whether the proposed framework performs well in practice compared to existing approaches.

- It would be helpful to summarize the existing results in a table, indicating which variants or contexts of the problem have been studied in prior work.

---

> ### Author Rebuttal · Authors · 2026-03-31
>
> We thank the reviewer for their thoughtful comments and suggestions.
>
> **Algorithmic Novelty.** To achieve the approximation guarantee for the cardinality constraint under both insertions and deletions—while maintaining constant consistency—our solution must simultaneously satisfy two properties. There must exist a threshold $\theta$ such that:
> - The average contribution of selected elements is at least $\theta$.
> - The marginal value of any non-selected element with respect to the solution is at most $\theta$.
>
> While this resembles the properties of the SIEVE streaming algorithm (Badanidiyuru et al.), there is a fundamental difference. In the standard streaming setting, it is sufficient to only satisfy one of the properties. Specifically, one needs the first property when the algorithm selects $k$ elements, and the second one when this is not the case.
>
> A main novelty of our work is to show that if we achieve both properties simultaneously at all times, then this implies good approximation in the fully dynamic setting. We show that this can be achieved by a novel algorithm that bridges greedy methods and threshold-based selection, both of which are staples of the submodular maximization literature in isolation. We combine this with the elegant sampling method of Zhang et al. to achieve robustness.
>
> Additionally, to achieve consistency, we design a novel RANDOM-SCHEDULING routine, which introduces two distinct conceptual novelties:
> - **Probabilistic Stability:** As correctly noted by the reviewer (RWMi), the random shift inherently ensures that any given timestep is a non-transition time with good probability.
> - **Robustness Convolution:** We introduce a convolution of different robustness levels. For every swap made in the matroid algorithm, we set an implicit expiration date determined by its robustness level (i.e., the size of the candidate element set). This nested structure guarantees that all active elements in the stream are eventually evaluated for eligibility, ensuring they are fully processed.
> This mechanism is vital for securing a constant factor approximation while restricting the consistency bound proportional to the number of robustness levels.
>
> We will clarify the relationship to Feldman, Karbasi & Kazemi (NeurIPS 2018) and investigate if the exchange-candidate procedure of FKK can be reused in our setting as a more generic and cleaner building block.
>
> **Role of consistency parameter C.** We clarify the role of the consistency parameter $C$. $C$ is not a data-dependent quantity which has to be estimated online, as per the reviewer comment, but a known parameter that solely depends on (i) the desired precision parameter $\varepsilon$ and - for matroids - (ii) on the rank $k$. The precision parameter $\varepsilon$ is set by the algorithm designer according to its requirement on the problem at hand, while the matroid rank $k$ is part of the input problem and known in advance.
>
> **Empirical Validation.** We agree that empirical studies are ultimately crucial. However, this paper establishes a novel frontier: we are the first to study submodular maximization subject to consistency constraints in the fully dynamic setting, handling both insertions and deletions. Because this introduces an entirely new problem formulation, the foundational algorithmic questions remain open. Our contribution focuses on establishing what is theoretically possible. Developing these necessary theoretical bounds is a standard first step in the community. Over the last few years, numerous ICML/NeurIPS papers focusing on submodular maximization in dynamic or online settings have been accepted solely on their theoretical merit, including:
>
> - "Uniform Wrappers: Bridging Concave to Quadratizable Functions in Online Optimization" (NeurIPS 2025)
> - "A Unified Approach to Submodular Maximization Under Noise" (NeurIPS 2025)
> - "Non-monotone Submodular Optimization: p-Matchoid Constraints and Fully Dynamic Setting" (NeurIPS 2025)
> - "A Near Linear Query Lower Bound for Submodular Maximization" (ICML 2025)
> - "From Linear to Linearizable Optimization: A Novel Framework..." (NeurIPS 2024)
> - "Gradient Methods for Online DR-Submodular Maximization..." (NeurIPS 2024)
> - "Dynamic Constrained Submodular Optimization with Polylogarithmic Update Time" (ICML 2023)
>
> We will add a clear statement in our introduction and conclusion explicitly defining this theoretical scope and noting empirical evaluation as an exciting direction for immediate future work.
>
> Table of Results. We thank the reviewer for the great suggestion. We will add to the camera ready a table summarizing our results and comparing them with the relevant results.

---

> > ### Author Rebuttal · Reviewer_XVSv · 2026-04-02
> >
> > Thank you for the detailed and thoughtful rebuttal. The explanations regarding the algorithmic design, the role of the consistency parameter, and the positioning of the work within the theoretical literature are helpful and address most of my previous concerns.
> >
> > In particular, the clarification on achieving both threshold properties simultaneously and the introduction of the RANDOM-SCHEDULING routine provide a clearer understanding of the technical contributions.
> >
> > However, I would appreciate further clarification on one key point: why is the consistency requirement fundamentally important in practice? While I understand its role in the theoretical formulation, the motivation for enforcing consistency—especially in comparison to standard dynamic or streaming settings—could be better articulated.
> >
> > Overall, I find the explanations satisfactory, and they address most of my concerns. Accordingly, I have increased my score to 4.

---

> > > ### Author Response · Authors · 2026-04-02
> > >
> > > We sincerely thank the reviewer for the thoughtful engagement with our rebuttal, for updating the score, and for raising a very pertinent question regarding the practical motivation for consistency. We agree that this is crucial, and we will explicitly expand upon it in the introduction of the final camera-ready version.
> > >
> > > We study fully dynamic submodular maximization with consistency constraints, extending prior work that has considered consistency in insertion only settings (Dutting et al. ICML 2024, Dutting et al STOC 2025). Bounding the worst-case number of changes per step is practically essential for several real-world reasons:
> > > - **User Experience and Cognitive Load:** In applications like recommendation systems, underlying catalogs evolve continuously. If retrieved suggestions completely change every time the dataset receives a minor update, it results in a jarring user experience. Consistency prevents these complete overhauls, ensuring that the retrieved solution remains relatively stable and predictable for the end user.
> > > - **Physical and Operational Costs:** In submodular applications like active learning or sparse reconstruction, changing the solution often incurs a tangible cost. Swapping an element might correspond to moving a physical sensor, allocating a new server, or triggering an expensive data-fetching operation. By strictly bounding the number of per-round updates, consistent algorithms directly limit these operational deployment costs
> > > - **Downstream Pipeline Stability:** Consistency can also be desirable in ML pipelines when downstream tasks rely on coresets or summaries. Relative stability of the summary allows downstream tasks to exploit caching and incremental updates.
> > >
> > > Standard fully dynamic algorithms focus primarily on minimizing *amortized* running time. To achieve this, they typically allow for the entire solution to be discarded and replaced periodically. While computationally efficient, replacing a whole solution is highly disruptive in practice. Something similar happens also in the streaming setting, where the goal is to maintain a *small summary* of the elements seen so far, which contains a high-value solution. Streaming algorithms for submodular maximization also naturally experience complete recomputation of the solution, thus violating consistency. Note, moreover, that it is impossible to achieve any constant factor approximation in the streaming setting when the stream also contains deletions.
> > >
> > > We hope this clarifies why bounding worst-case changes is often just as critical as bounding running time or memory footprint in real-world deployments.

---

### Official Review · Reviewer_x18Q · 2026-03-13

**Soundness:** 4
**Presentation:** 4
**Significance:** 3
**Originality:** 3
**Overall Recommendation:** 5
**Confidence:** 4

**Summary:**

The paper studies fully dynamic consistent monotone submodular maximization under either a cardinality constraint or matroid constraint. At each time $t$, the stream performs either an insertion or a deletion of an element $x_t$, the active set is $X_t$, and the algorithm must maintain a feasible solution $ALG_t \subseteq X_t$ that simultaneously satisfies an approximation guarantee $\mathbb{E}[f(ALG_t)] \ge \alpha \cdot, f(OPT_t)$ and a worst-case per-step consistency bound $| ALG_t \Delta ALG_{t-1}| \leq C$. The paper replaces the insertion-only recourse metric used in earlier work with symmetric-difference consistency, which is the natural notion once deletions are allowed.

The main contribution is a general modular framework built from three components: a randomized multi-scale scheduler a.k.a. Random-Scheduling, robust submodular routine $\mathcal{A}_R$, and a non-robust routine $\mathcal{A}_N$. All the moving parts of the framework are non-trivial.

Using the above framework, forr \emph{matroid constraints}, the algorithm Consistent-Matroid achieves a polynomial-time $\frac 1 4$-approximation with $O(\log k/\varepsilon^2)$ consistency. For \emph{cardinality constraints}, the framework becomes both simpler and stronger: they obtain polynomial-time $1/2$-approximation with $O(1/\varepsilon^2)$ consistency.

Overall, the key technical contribution is the first general recipe that turns deletion-robust submodular primitives into fully dynamic, worst-case consistent algorithms, giving the first constant-factor guarantees with consistency sublinear in $k$ in this model.

**Compliance With Llm Reviewing Policy:**

Affirmed.

**Key Questions For Authors:**

Do you believe the algorithms you provided could lead to sublinear update time for dynamic data structure for maximizing submodular objective subject to cardinality and more generally matroid constraints?

**Limitations:**

"yes"

**Strengths And Weaknesses:**

I enjoyed reading the paper, the result itself crosses the bar of ICML. They obtain the first sublinear consistency for submodualr maximization in the fully-dynamic setting. Even though approximation ratios are not tight, the authors make a significant step in the direction of generalizing the result of Duetting et. al. STOC 2025 which gives (randomized) algorithm for increasing stream. The result for cardinality constraint almost matches the guarantees for increasing stream form Duetting et. al. STOC 2025.

One weakness I see is that the paper focuses on consistency rather than dynamic running time, so the algorithms are only stated as polynomial-time. For evaluating practical impact, it would help to know the oracle complexity per update and how expensive the repeated robust recomputations are compared with existing fully dynamic algorithms that allow large recourse.

---

> ### Author Rebuttal · Authors · 2026-03-31
>
> We thank the reviewer for their thoughtful comments and suggestions. We comment here on the difficulty of achieving sublinear update time.
>
> The main bottleneck in achieving sublinear update time with our framework resides in the fact that every $\approx k$ time steps, we need to process all the elements from scratch, for an overall update time that is $\approx n^2/k$. To achieve sublinear update time, we would need to decrease the frequency of such recomputations, without affecting the consistency guarantees.

---

> > ### Author Rebuttal · Reviewer_x18Q · 2026-04-02
> >
> > I keep my score unchanged.

---

### Official Review · Reviewer_RWMi · 2026-03-16

**Soundness:** 3
**Presentation:** 3
**Significance:** 3
**Originality:** 2
**Overall Recommendation:** 3
**Confidence:** 3

**Summary:**

The paper studies fully dynamic monotone submodular maximization under a
consistency constraint: the algorithm must maintain a near-optimal solution
while changing at most C elements per update (insertion or deletion). Prior
consistency results (Dutting et al., ICML 2024; STOC 2025) handled only
insertion-only streams; prior fully-dynamic results (Lattanzi et al.,
NeurIPS 2020; Chen & Peng, STOC 2022) bounded running time but not
solution changes. This paper fills the intersection: fully dynamic +
consistent + constant-factor approximation.

Results:
- Cardinality constraint: (1/2 - O(ε))-approximation, O(1/ε²)-consistent
- Matroid constraint (rank k): (1/4 - O(ε))-approximation, O(log k / ε²)-consistent

**Compliance With Llm Reviewing Policy:**

Affirmed.

**Final Justification:**

The authors partially addressed my concerns -- they provided the query complexity computation, which I think would be a valuable addition to the paper.

My concern about the theoretical novelty remains. However, I do not feel strongly about the paper and would be fine if it were accepted.

Thus, my score will be left unchanged.

**Key Questions For Authors:**

1. What is the amortized or worst-case per-step oracle query complexity
   of your algorithms? If the query complexity is polynomial in k and
   1/ε, please state the bound explicitly and compare it to prior
   fully-dynamic algorithms. If consistency introduces additional query
   overhead compared to non-consistent fully-dynamic algorithms, please
   quantify the tradeoff.

2. The ROBUST-SWAP subroutine uses the same circuit-based exchange
   condition as Chakrabarti & Kale (IPCO 2014) and Feldman, Karbasi &
   Kazemi (NeurIPS 2018, EXCHANGE-CANDIDATE). Can you clarify what, if
   anything, is algorithmically new in ROBUST-SWAP beyond the
   inverse-marginal sampling (which is from Zhang et al. 2022) and the
   candidate-count filter? If the novelty is entirely in the
   RANDOM-SCHEDULING wrapper and the consistency analysis, that should
   be foregrounded.

3. Can you provide even a simple synthetic experiment demonstrating a
   regime where the consistency guarantee provides practical benefit over
   a "rebuild from scratch" baseline? If consistency is motivated by
   practical concerns, the paper should demonstrate that it delivers
   practical value.

**Limitations:**

yes

**Strengths And Weaknesses:**

## Strengths

- The problem formulation is well-motivated. Fully dynamic + consistent is
  a genuine gap in the literature, and the paper correctly identifies that
  deletions break the monotonicity-of-OPT property that prior consistency
  papers exploit.

- The RANDOM-SCHEDULING mechanism (hierarchical transition scheme with
  uniform random shift) is clean and potentially reusable. The idea that
  any given step is a "transition time" with probability at most ε is
  elegant.

- The paper is clearly organized. The framework structure (abstract
  subroutines + scheduling wrapper) makes the approach easy to follow.

- The approximation ratios (1/2 - ε for cardinality, 1/4 - ε for matroid)
  match the best known fully-dynamic ratios without consistency constraints,
  suggesting consistency may come "for free" in terms of approximation
  quality. This would be a strong result — but see Weakness 2 on why we
  cannot verify this claim.

## Weaknesses

### 1. The core algorithmic components are well-established, not new

The paper's two main subroutines closely follow techniques from over a
decade of streaming and dynamic submodular maximization:

**ROBUST-SWAP (Algorithm 1, matroid constraint):** For each candidate u,
compute exchange candidates Γ_u = {k ∈ I | I − k + u ∈ M}, select the
minimum-weight k_u, and swap if f(u|I) ≥ 2·w(k_u). This is the
Chakrabarti-Kale (IPCO 2014) swap condition with γ = 1 (since 2 = 1+γ).
The circuit-based exchange-candidate identification is the same procedure
that Feldman, Karbasi & Kazemi (NeurIPS 2018, "Do Less Get More") explicitly
isolated as a named subroutine (their Algorithm 1, EXCHANGE-CANDIDATE) and
proved correct for the strictly more general setting of non-monotone
objectives over p-matchoid constraints. The inverse-marginal-contribution
sampling is from Zhang et al. (2022).

**ROBUST-GREEDY (Algorithm 3, cardinality constraint):** Greedy insertion
with a marginal-gain threshold filter. This follows the template of
Badanidiyuru, Mirzasoleiman, Karbasi & Krause (KDD 2014, Sieve-Streaming),
which introduced threshold-based filtering for cardinality-constrained
streaming submodular maximization. The addition of inverse-marginal
sampling from the top candidates is again from Zhang et al. (2022).

The split between "threshold-add for cardinality, exchange-swap for
matroids" has been the standard template in this area since 2014. The
paper follows it exactly. The only genuinely new component is
RANDOM-SCHEDULING — which is an organizational device (a scheduling and
analysis wrapper), not an algorithmic technique. It is clean work, but
it is analysis rather than algorithm design.

The paper should discuss its algorithmic lineage more transparently. In
particular, the relationship to Feldman, Karbasi & Kazemi (NeurIPS 2018)
deserves explicit acknowledgment in the algorithmic sections, as FK
isolated the exchange-candidate procedure as a reusable primitive and
proved it in a more general setting.

### 2. Per-step query complexity is not analyzed

The paper bounds the number of solution changes (swaps) per update but
does not state a per-step oracle query complexity. Prior fully-dynamic
papers explicitly bound computational cost:

- Lattanzi et al. (NeurIPS 2020): amortized query complexity per update
- Chen & Peng (STOC 2022): amortized query complexity per update
- Dutting et al. (ICML 2023): amortized query complexity per update

Without this analysis, we cannot determine whether consistency comes at
a computational cost. There are three possible realities: (a) consistency
is free — the per-step query complexity matches prior fully-dynamic
algorithms; (b) consistency costs something quantifiable, e.g., an extra
factor of 1/ε — which is a natural tradeoff, perfectly acceptable to
disclose; (c) the authors have not analyzed it. Any of these is fine to
state. What is not acceptable is omitting the analysis entirely, because
it prevents the reader from comparing this algorithm's total
computational cost to prior work.

If consistency introduces overhead, that is not a weakness of the
algorithm — it is a tradeoff inherent to the problem. But it must be
quantified, not swept under the rug.

### 3. No experiments despite a practical motivation

The paper motivates consistency with practical considerations: rebuilding
solutions from scratch is disruptive in applications like recommendation
systems, sensor placement, and data summarization. This is a reasonable
motivation — but the paper then provides no experimental evidence that
the consistency guarantee matters in practice.

For a theory paper at ICML (as opposed to STOC/FOCS/SODA), either the
techniques must be genuinely novel or the results should be validated
experimentally. This paper's techniques are largely known (see Weakness 1),
which makes the experimental gap more significant.

A compelling experimental section would: (a) demonstrate a practical
regime where consistency provides meaningful benefit (e.g., reduced
deployment cost or user disruption); (b) show the approximation quality
vs. consistency tradeoff empirically; and (c) honestly characterize
where the algorithm's consistency guarantee does and does not help. Such
experiments would substantially strengthen the paper.

### 4. The "general framework" claim is oversold

The paper claims a general framework but provides only two instantiations:
cardinality and matroid constraints. The open problems section acknowledges
that extensions to non-monotone objectives and knapsack constraints are
open — suggesting the framework does not trivially extend. "Framework" is
aspirational; "design pattern instantiated twice" is more accurate.

### 5. The 1/4 ratio for matroids may be improvable

The 1/4 factor for matroid constraints appears to come from the
weight-based analysis losing a factor of 4 (the standard CK analysis
artifact with γ = 1). The best offline matroid-constrained submodular
maximization ratio is 1 − 1/e ≈ 0.632 (Calinescu et al.). Whether the
1/4 is a fundamental barrier for the consistent dynamic setting or a
proof artifact is worth discussing. If the latter, it weakens the paper's
claim of achieving "the best possible" ratios.

---

> ### Author Rebuttal · Authors · 2026-03-31
>
> We thank the reviewer for their thoughtful comments and suggestions.
>
> **Update Time.** We thank the reviewer for the opportunity to clarify the computational complexity of our algorithms. In this setting, there are three natural, competing objectives: (i) approximation guarantees, (ii) consistency, and (iii) optimal update time. We believe it is conceptually cleaner to first isolate and understand the fundamental tradeoff between the first two (which has been called the ``cost of consistency’’ in prior work). Therefore, while we ensure our algorithms maintain a worst-case amortized running time that is polynomial, simultaneously optimizing for the tightest possible update complexity was intentionally left outside our primary theoretical scope.
>
> We will gladly add a formal running time analysis, derived directly from the pseudocode, to the final version. As is standard in the submodular literature, we measure complexity via the number of calls to the value and independence oracles. Scheduling operations do not query these oracles, so we focus on the online operations. Omitting lower-order terms, the polynomial dependencies on $n$ and $k$ are as follows:
>
> - For matroids the worst-case amortized running time is $O(n^2/k + k^2)$.
> - For the cardinality case the amortized update time is $O(n)$.
>
> Note, if we were to optimize update time, one could likely use some “lazy update” trick (à la Minoux, see e.g., “Accelerated greedy algorithms for maximizing submodular set functions” in Optimization Techniques, 1978) to implement RobustGreedy and RobustSwap with less update time, however this would not be enough to get sublinear update time (we discuss the technical bottlenecks that currently prevent us from achieving sublinear update time in our reply to reviewer x18Q).
>
> **Algorithmic Novelty.** Please see our reply to the Reviewer XVSv.
>
> **Empirical Validation.** Please see our reply to the reviewer XVSv.
>
> **Generality of our framework.** To clarify the scope of our contribution, we used “general framework” to highlight the unified algorithmic "recipe" used to derive our results. In this work, we use this recipe to achieve fundamental new results for cardinality and matroid constraints, which are well established benchmark problems in this area. The same recipe should be applicable more broadly. For instance, we could use the general version of the algorithm by Chakrabati and Kale to address $p$-matroids or more general constraints. Similarly, we suspect that adding a sub-sampling step (e.g., Buchbinder et al. “Submodular maximization with cardinality constraints.”, SODA 2014) to our current algorithm may directly achieve constant factor approximations for non-monotone functions (this procedure has already worked seamlessly in the related deletion robust setting, see the JMLR version of Duetting et al. ICML 22).
>
> **On the tightness of approximation ratios.** Our results are not tight with respect to approximation ratio, and may be improved in future work, possibly up to the offline $1-1/e$. However, we point out that there are many situations where this is known to be impossible (e.g. in the streaming model). For other lines of research on submodular maximization, it took years (if not decades) to reach tight results while many fundamental problems still remain open. Beyond the open gaps discussed in the last paragraph of Our Results, two additional examples that we discuss in the related work are:
> - The STOC 25 papers by Duetting et al. (following up on an ICML 24 paper) on insertion-only consistency with cardinality constraints achieves tight information theoretical bounds, but leaves open a gap for polytime algorithms $(0.51, 1 − 1/e]$.
> - The state of the art for fully-dynamic submodular maximization with matroids constraints (Banihashem et al., SODA 2024; Duetting et al., TALG 2025) exhibit the same $¼$ approximation result, which is still far from the upper bound of $½$ (which holds also for cardinality).

---

> > ### Author Rebuttal · Reviewer_RWMi · 2026-04-02
> >
> > The authors partially addressed my concerns -- they provided the query complexity computation, which I think would be a valuable addition to the paper.
> >
> > My concern about the theoretical novelty remains. However, I do not feel strongly about the paper and would be fine if it were accepted.
> >
> > Thus, my score will be left unchanged.

---

### Decision · Program_Chairs · 2026-04-30

**Decision:**

Accept (regular)

**Comment:**

The paper studies the problem of dynamic consistent submodular maximization, where the ground set evolves over time through insertions and deletions, and the algorithm must maintain a near-optimal solution under a consistency constraint.

Most reviewers acknowledged the value of this work: well-motivated problem formulation, offering non-trivial algorithmic insights for submodular maximization in the fully dynamic setting, giving the first constant-factor guarantees with consistency sublinear in this model. Some reviewers also raised the concerns on the novelty: the core algorithmic components are well-established, and the technical analysis mainly relies on standard techniques. Overall, reviewers generally agree that this paper provides meaningful contributions.